# Exploring Inpatients’ Perspective: A Cross-Sectional Survey on Satisfaction and Experiences in Greek Hospitals

**DOI:** 10.3390/healthcare12060658

**Published:** 2024-03-14

**Authors:** Dimitris Charalambos Karaferis, Dimitris A. Niakas

**Affiliations:** Department of Health Economics, Medical School, National and Kapodistrian University of Athens, 11527 Athens, Greece; dimitris.niakas@gmail.com

**Keywords:** patients’ satisfaction, healthcare quality improvement, public hospitals, Greece

## Abstract

Introduction: The aim of this study was to identify and evaluate patient-relevant experiences that fulfill the expectations and demands of society in Greece and those that could be improved by offering a better quality of care. The satisfaction of health service recipients is one of the key elements of the success of a health system. Methods: A cross-sectional survey was conducted to obtain data on satisfaction with hospitalization from patients admitted to 10 public hospitals in Athens between June 2019 and December 2021. Statistical analysis was applied to 57 items and 7 dimensions of patient satisfaction, namely waiting–arrival–admission, nursing staff, medical staff, other staff, service and quality of food, interior environment, and procedures. Results: A total of 3724 patients, aged ≥ 18 years, who had experienced hospitalization and agreed to participate in the study were included, the response rate of which was 93%. Patient satisfaction and experience with healthcare services provided by hospitals is moderate, with almost two-thirds of patients (67.38%) satisfied with the care they received. The encounter with the medical–nursing personnel (3.75/5) and other staff (4/5) were factors that positively affected patients’ overall satisfaction with hospitalization. However, there were some causes of dissatisfaction, mainly associated with waiting hours, easy access to medical services or services received in emergencies, delays of planned procedures (3.50/5), or problems with old facilities and equipment (3.56/5). Conclusion: Based on the patients’ judgment, the performance of hospitals was rated at a ‘tolerable’ level. Professionalism and the education of personnel led to a positive treatment outcome and improved the experience of patients to a good level. However, public hospitals continued to be underfunded and lacked strong support, which affected staff communication and responsiveness to patients’ requirements, while smart technologies and the simplification of procedures were not adopted to help staff provide a better quality of healthcare. The results suggest that there is plenty of room for improvement.

## 1. Introduction

Unfortunately, in the past, the quality of and improvement in healthcare services has focused on factors that professionals believe should be evaluated, paying less attention to factors that patients consider important or not directly seeking patients’ views. As a result, data were sparse and underutilized, and methodological problems and a lack of comparability across countries have led to gaps in the literature on patients’ perspectives [1]. Nowadays, a more pluralistic approach has been adopted by evaluating multiple perspectives. This approach to evaluation can transcend professionally dominated traditions of healthcare evaluation by identifying and representing the views of the stakeholder group, including the marginalized [2]. In simple terms, patient satisfaction is how well patients are treated, and ‘how well’ refers not only to the quality of care, but also to how satisfied patients are with the treatment they receive. ‘Patient experience’ and ‘patient satisfaction’ are often used interchangeably in healthcare, but they are not identical concepts. Patient experience is based on what should happen during the patient’s stay and whether it actually happened, a concept related to testing a defined quality of standard of care. Meanwhile, patient satisfaction is very personal and is based on whether the patient’s expectations of what should happen were met. So, this disparity is a case of subjectivity versus objectivity [3,4].

More recently, patient satisfaction has become a highly desirable outcome in terms of the quality and cost of healthcare services and an even more important indicator of patient safety. Studies have shown that hospitals with higher patient satisfaction scores achieve better adherence to treatment regimens, receive fewer complaints, and perform better on quality indicators [5,6]. While the goals of safety and effectiveness in healthcare are largely universal, the extent to which additional goals such as access, efficiency, timeliness, evidence-based treatment, equity, and patient-centeredness are emphasized varies across healthcare systems, societies, and cultures around the world [7]. Additionally, interest in healthcare has been strongly recognized and accelerated lately, as people’s demands for a healthier lifestyle have intensely increased and the trends of the population have shifted toward attaining a healthier lifestyle [8]. The advent of the Internet and social media platforms has given patients the opportunity to learn more about the quality of care that healthcare providers can offer. Approximately, it is believed that during a typical 3- to 4-day stay in a hospital, a patient may interact with 50–60 employees [9]. Bearing in mind the above, patients are able to set new expectations for convenience, transparency, collaboration, and healthcare facilities, forcing healthcare providers to develop strategies to meet these new demands [10].

Quality of care is a dominant concept in quality assurance and improvement programs, since hospitals’ efforts to satisfy patients are directly linked to key success metrics and are fully aligned with the delivery of individualized medical services. Consequently, improving patient-centered care has become a priority for all healthcare providers to achieve the goal of high patient satisfaction. Patients’ expressions of satisfaction or dissatisfaction assess all aspects of hospital care quality. A loyal patient may not only comply with medical treatment but will also recommend that hospital to other patients in need. On the other hand, a disloyal patient may no longer use that hospital’s healthcare services and change healthcare providers or physicians [11,12]. Many researchers insist that the most realistic approach to patient satisfaction is to measure patients’ perceptions directly by using a questionnaire method, as this multi-dimensional concept includes both medical and non-medical aspects of healthcare. Moreover, patient satisfaction researchers argue that attribute-based ‘satisfaction’ assessments provide elements that help identify tangible priorities for quality improvement and a richer measure for several reasons: (a) they allow for a better understanding of how patients perceive hospital operations, treatment modalities, and administrative aspects, (b) they ensure the implementation of best practices that deliver better results, (c) the levels of satisfaction are directly linked to the community and its use of the service, and (d) they act as a mean of feedback. Moreover, without collecting and monitoring patient satisfaction data, hospitals cannot know how well they are performing. Of note, given that healthcare services are provided free of charge by the government, public spending on healthcare has a significant impact on patient satisfaction. Thus, opinions based on the judgments of patients who are taxed and pay for health services are more important for highlighting problems within the institution and justify an assessment of its level of functioning [13,14,15,16].

Previously published studies in Greece have shown that understaffing, budget cuts, and low decisional control of patients are dimensions that contribute to patients’ dissatisfaction in hospitals. Negative gaps also exist in other areas, namely safety, empathy, reliability, and responsiveness, and these dimensions refer primarily to interpersonal relationships, transactions, and encounters between patients and professionals. Negative gaps in these dimensions indicate deeper problems with hospital quality of care [17,18,19,20,21,22]. Hence, Greece has one of the lowest overall response rates for inpatient services among OECD countries and the healthcare system has been strongly hospital-centered due to transformations and reforms in the primary care system during 2011–2018 [23,24,25]. In this light, and since patient satisfaction is not a static indicator as it is affected by changes in the external environment, this study aims to provide valuable insights, identify areas that require attention, assess the challenges associated with enhancing patient satisfaction, and bring to the forefront key concepts that could enhance the quality of care in hospital settings.

## 2. Materials and Methods

### 2.1. Instrument

This study is based on the Aletras, Basiouri, Kontodimopoulos, Ioannidou, Niakas (2009) questionnaire. Their survey was conducted with a sample of 150 inpatients hospitalized in April 2005 in a general public hospital in Veroia, Greece. The response rate was 64.65%. Exploratory factor analysis was conducted to explore factors within the questionnaire items and to verify the reliability and validity of the service dimensions on patients’ satisfaction. Principal components analysis revealed four factors, namely ‘physician and nursing care’, ‘organization of care’, ‘hospital environment’, and ‘other quality factors’, that had mean (median) satisfaction scores of 4.36 (4.42), 4.59 (4.83), 4.26 (4.22), and 4.49 (4.50), respectively. Cronbach’s coefficients took values between 0.85 and 0.96. Test–retest reliability coefficients were between 0.814 and 0.970, whereas correlations of inter-rater reliability ranged from 0.811 to 0.978. [26].

The questionnaire of this study was presented to the patients in two parts. Ιn Part A, we tried to retrieve information from patients’ experiences in the hospitals of the 1st Health Region of Attica. Initially, a question was posed regarding the mode of admission to the hospital, whether it was, in other words, in an emergency or scheduled through a waiting list; this was followed by 56 questions, most of which are closed Likert-type questions, with a 6-point intensity scale ranging from strongly disagree to strongly agree, and taking numerical values: 1 = strongly agree, 2 = agree, 3 = neutral, 4 = disagree, 5 = strongly disagree, and 6 = don’t know/no answer. Included in the survey are 12 questions (2–13) related to waiting–arrival–admission, 9 questions (14–22) related to the behavior of nursing staff, 9 questions (23–31) related to the behavior of medical staff, and 5 questions (32–37) related to the rest of the hospital staff. In addition, there are 3 questions (38–40) about the food situation, 10 questions (41–50) referring to the internal environment, 4 questions (51–54) about the planning of hospital procedures, and 3 questions (55–57) about the overall evaluation of the hospital. The questionnaire includes questions with statements, either positive or negative, so as to exclude positive bias. The final questionnaire, in total, includes 58 questions of which questions 2, 4, 6, 8, 10, 15, 17, 19, 21, 23, 25, 27, 29, 31, 33, 34, 38, 40, 42, 44, 46, 48, 50, and 52 are reversed before the statistical analysis, since they are worded in an opposite way to the others. Part Β includes sociodemographic data, like patients’ gender, age, marital and educational status, professional category, nationality, and economic situation.

### 2.2. Ethical Permission

The Ethical Committee of the National and Kapodistrian University of Athens approved the study protocol (1819023327-25/2/2019). Approval was also obtained from the 1st Regional Health Authority of Attica (approval number: 31707-7/6/2019). The survey was conducted between June 2019 and December 2021 in 10 hospitals out of a total of 24 in the 1st Regional Health Authority of Attica; we obtained written approval from competent institutional ethics and research committees from all hospitals, provided that the hospitals are not presented by name. The research process is consistent with the Helsinki Declaration of 2013 [27].

### 2.3. Participants and Procedure

The Attika region is the largest region of Greece, with a total population of around 3.75 million, approximately 35% of Greece’s total population. According to the Greek Ministry of Health, the Attica region, across a total of 24 General Public Hospitals, had created 8631 hospital beds and provided health services to 589,448 patients in 2021. In the same year, the ten hospitals that we studied had created 5313 hospital beds (62% of total) and provided healthcare services to 392,425 patients (67% of the whole access population) [28].

The research design is a cross-sectional analysis and the sampling technique used was convenience sampling. This is a non-probability sampling method and samples are drawn from a group of patients who are easy to contact or reach (i.e., those who were more willing to participate in the survey).

The patients were informed of their participation in the collection of information to assess the quality of hospital care. The survey was anonymous and voluntary, and the patients had the option not to provide their opinions. The questionnaires were obtained from the patients by the researchers themselves, and in some cases, a relative was present, after the method and time allowance for completing the questionnaires as well as the importance of their participation in the study were explicitly explained to them. Patients had to be at least 18 years old, know the Greek language, be able to communicate and speak with other people, and have stayed in hospital for at least 24 h.

### 2.4. Statistical Analysis

Descriptive statistics were used to report respondents’ level of patient satisfaction, including means, standard deviations, and median (interquartile range). Also, the categorical variables are presented as absolute (N) and relative (%) frequencies. The Kolmogorov–Smirnov test was used for normality assessment. To compare the overall satisfaction score as a quantitative variable between three or more different groups, we used the ‘ANOVA’ method and a non-parametric test, the ‘Kruskal–Wallis’ test, while for comparing the overall satisfaction score between two groups, we used the ‘*t*-test’ and non-parametric testing of ‘Mann–Whitney’. The individual’s responses to the “don’t know/don’t want to answer” options in the questionnaire were not included for the evaluation of acceptability [26,29,30]. Reliability analysis included Cronbach’s alpha for internal consistency. Subsequently, multiple linear regression models were used to predict the factors associated with patients’ satisfaction with healthcare service quality. Measures expressed as coefficient beta (β) with a 95% confidence interval were used to describe the association among the variables. Statistical significance was set at a two-sided *p*-value < 0.05. All statistical analyses were performed using SPSS 26.0 (IBM Corp., Armonk, NY, USA).

## 3. Results

### 3.1. Normality and Reliability Analysis

Kolmogorov–Smirnov and Shapiro–Wilk tests were applied to assess the normality of the distributions. These tests showed a statistically significant deviation from normality. Internal consistency and the reliability of the satisfaction questionnaire were analyzed, and Cronbach’s alpha coefficient was found to be 0.75 [31].

### 3.2. Analysis of Sociodemographic and Hospitalization Characteristics of Study Participants

A strong feature of this study is the large sample size. A total of 3724 adult patients took part in this research and were included in our analyses. The response rate was 93% (3724/4000). The structure of the research sample is presented in Table 1.

By investigating the medical profile of the interviewed patients in relation to the factors specifically mentioned in their hospitalization, it is shown that 60.23% (Ν = 2243) of the total hospitalizations were emergencies, while 39.77% (Ν = 1481) were admitted as preplanned. In relation to gender, among men, 29.99% were admitted in emergencies, while 18.39% of admissions were preplanned. Of the women, 30.24% were admitted in emergencies, while 21.37% of hospitalizations were preplanned. The number of patients who were admitted after an appointment, i.e., scheduled, were (Ν = 1481). Of these, 5% (N = 74) of the sample were waiting for less than 1 week, a 1- to 2-week wait was experienced by 6.62% (N = 98), and between 2 and 4 weeks of waiting was the case for 36.66% (N = 543) of the sample. The largest percentage of the sample, 49.29% (N = 730), had to wait to be admitted for 1–2 months. After their admission, the majority of surveyed patients, 55.13% (N = 2053), stayed in the hospital clinics for between 6 and 15 days, 41.19% (N = 1534) for between 2 and 5 days, while the remaining 3.69% (N = 137) stayed for more than 16 days. In a question asked about the waiting period for admission onto a ward, 41.68% (N = 1552) of patients stated a wait from 31 min to 1 h, 30.21% (N = 1125) a wait of 1–2 h, while 2 out of 10 stated a wait of more than 2 h.

### 3.3. Patients’ Measurement of Satisfaction and Experiences

Systematic gathering of information on patients’ needs and experiences, using methodologically sound instruments, should be an integral part of routine care. In this study, a structured and validated questionnaire was used, appropriately adjusted to the needs of the study to gather information about patients’ reported experiences [26]. In Table 2 below, the satisfaction rate and descriptive statistics of the research sample are presented.

### 3.4. Scores of Dimensions Associated with Satisfaction of the Admitted Patients

Exploratory factor analysis was used to classify the questionnaire and define four dimensions, namely (1) medical treatment and nursing; (2) organization and planning of enrollment; (3) hospital environment; (4) other qualitative factors. Table 3 below provides details of patient responses and related questions for each dimension [26]. The two dimensions of ‘hospital environment’ and ‘organizational processes’ clearly did not meet patients’ expectations.

### 3.5. Multiple Linear Regression Analysis of Predictors Associated with Patient Satisfaction

The multiple linear regression analysis was employed to identify the determinants that have the most significant impact on patient satisfaction in public sector hospitals. From the result of the coefficient in Table 4, it can be understood that ‘organization and planning of hospitalization’ and ‘hospital environment’ are the strongest and have a positive and significant relationship with patient satisfaction (β = 0.215; *p* < 0.001), being predictors for satisfaction. Furthermore, ‘other quality factors’ also have a positive and significant correlation with (β = 0.207; *p* < 0.001) patients’ satisfaction, followed by ‘medical and nursing care’ (β = 0.175; *p* < 0.001).

### 3.6. The Effect of Sociodemographic Variables on Patients’ Satisfaction

An analytical study was carried out to obtain the possible associations between the sociodemographic variables of the patients and the level of satisfaction achieved, finding the existence or not of statistically significant differences. Table 5 describes the rates of the overall satisfaction scores regarding the healthcare service depending on sociodemographic and other factors.

The results show that most of the sociodemographic factors (such as education level, economic situation, and residence) and patient satisfaction had no statistical differences (*p* > 0.05); however, gender, marital status, and nationality were significant. Moreover, it is observed that male patients reported higher overall satisfaction than female patients.

### 3.7. Reasons Related to Preferring and Recommending a Hospital, and Generally Favorable Thoughts about the Services Provided in Hospitals

Table 6 shows the main influencing factors that drive patients to develop a preference for and thus recommend a hospital, and to generally have positive feelings and thoughts about the services provided.

Multivariate linear regression revealed that the interest, polite behavior and communication, and empathy of medical and nursing personnel offering care and information, the waiting times, a trustful atmosphere throughout the hospital, scheduled procedures, and the situation of the ward environment were the key influencing factors related to preferring and recommending a hospital.

## 4. Discussion

The results of our analyses revealed interesting findings about the services provided in hospitals. Τhe overall satisfaction of hospitalized patients was 67.38%. Only 52% (N = 1929) of patients expressed satisfaction with wait–arrival–admission. The waiting time of those who came by appointment took more than a month for their case to be processed. Additionally, emergency cases and diagnostic results showed long delays, since it was reported that the admission time until a bed was found in a ward for 30.21% (N = 1125) of patients was between 1 and 2 h, while 23.15% (N = 862) reported a 2 h wait or even longer. Positive comments were made about the efforts of the staff in the registration and admission office. Also, it was stated that people with disabilities had accessibility.

The responsiveness of the nursing staff was rated positively, and patients were satisfied with their courtesy (97.07%), respect (95.03%), discretion (96.91%), interest (84.34%), and willingness to answer questions (72.82%); however, it was stated that the number of nursing staff was insufficient to cover the existing needs of patients (83.36%), and this fact affected the development of interpersonal relationships between nursing staff and patients. Empathy (44.42%), explanations of procedures (33.47%), and responses to questions (51%) were rated low, which suggests that communication between staff and patients needs to be improved as nursing staff are professionals who are not only in direct contact with patients but can also psychologically support patients and their families. Many authors highlight that the interpersonal behavior of employees (interest, friendliness, empathy, courtesy, respect for treatment preferences, and communication with family) influences overall satisfaction with health services [32].

The majority of patients agreed that physicians have a polite manner and discreet behavior (98%) when providing care. A significant percentage of patients expressed satisfaction with the provision and completeness of the instructions given (91%) and with the information they had about their health (94.23%). Research has shown that hospitals in this region have a high level of educated personnel [33]. Regarding the availability of physicians, only 64.57% expressed satisfaction and a high proportion of patients (67%) reported that their needs were not adequately met. The physician–patient relationship is particularly important as it provides a therapeutic aspect; this relationship requires connection, respect, compassion, and reflection. Evidence suggests that satisfied patients are more likely to use services and continue their treatment, while dissatisfied patients are at greater risk of discontinuing treatment [34,35,36]. The international literature highlights the importance of the interpersonal aspect, communication, and professionalism of healthcare in shaping patient satisfaction [37,38,39]. It is estimated that 70–80% of legal referrals or errors made by health professionals are due to problems in the relationship between health professionals and patients, information problems, and poor teamwork or communication [40,41,42,43,44].

As for the rest of the hospital staff, namely those who had undertaken the feeding services, the patients expressed satisfaction at a high rate of 94.69%, and they expressed satisfaction in the same way for the behavior of the cleaning staff (91.92%). Regarding the food quality, the patients expressed satisfaction at a rate of 64.10%, as the food may have been poorly served or not particularly tasty. However, the distribution of meals was achieved within the time frames.

The internal environment includes all of the situations experienced in the ward during patients’ hospitalization. Concerning the cleanliness of the ward, 79.87% expressed satisfaction. For the calmness of the wards, the patients expressed satisfaction at a rate of 79.62%. In most cases, those who expressed dissatisfaction were in smaller wards with more beds. It was also noted that there is no special care for attendants, as a result of which they are forced to spend periods in uncomfortable chairs or take special armchairs privately. In addition, the attendants complained about the organization and lack of maintenance of the waiting areas outside the chambers, as well as the existence of insufficient and broken furniture. Referring to the changing of bed linen, only a small percentage (17.25%) expressed satisfaction, with most of the patients reporting using their own bedding. Regarding the ideal temperature in the wards, 86% expressed satisfaction. Examining cleanliness in the sanitary areas, bathrooms, and toilets, 75.34% expressed satisfaction. Many attendants indicated that there was a serious problem of a lack of toilets within the wards or not having many toilets on each floor. Also, they indicated that there was an insufficient supply of materials (detergents, antiseptics, soaps, and toilet paper) and highlighted the presence of many insects. The age of the buildings was also cited as a cause of inconvenience, with patients arguing that repairs were needed and renovation was necessary. When patients were asked if they felt uncomfortable because there were more patients on the ward, the majority (77.32%) said there was no problem. Regarding the security that patients felt about their belongings in case of possible theft, only 53.29% expressed satisfaction. In the question “Did you feel uncomfortable because the relatives and friends of the patients stayed in the ward for a long time”, 65.60% of the patients stated that they did not have a problem, but many people complained that visiting hours are not respected. It is already known that there is a close relationship between noise levels in hospital rooms and patient satisfaction [45]. Regarding the cleanliness of the rest of the hospital’s waiting areas (e.g., corridors, canteen, offices, and waiting areas), 70.82% stated that they were clean. Concerning the planned procedures, 68.05% expressed satisfaction. To measure patients’ attitudes toward the hospitals they attended, we used two additional related questions (Q56–57). With the question Q56, “Would you choose to come to the Hospital again?”, the patient’s intention to revisit the hospital if necessary was estimated, and a high percentage of patients (89.23%) agreed that they would return. Furthermore, with the question Q57, “Would you recommend the Hospital to your friends, family and third parties?” the patient’s intention to recommend the Hospital to other third parties or to their own family and friends was evaluated, and 69.16% answered positively. Given that repeated satisfying experiences with care providers build trust, which in turn leads to loyalty, it is undeniable that trust can influence satisfaction [46,47].

In Greece, the ongoing economic crisis (2009–2019) affected the overall performance and staffing of hospitals. Reducing expenses for healthcare reduced the quality of care. Additionally, 34% (2017) of health expenditure was financed by households, remaining among the highest in the EU, and 8.8% stated that their “needs were not met”, compared to the average of 1.8% of the EU. In this survey, the four dimensions, namely ‘physician and nursing care’, ‘organization of care’, ‘hospital environment’, and ‘other quality factors’ had mean (median) satisfaction scores of 3.73 (3.75/5), 3.35 (3.50/5), 3.44 (3.56/5), and 3.80 (4.00/5), respectively. Our findings are in line with recent studies that highlight communication, a trustful atmosphere throughout the hospital, individualized attention, responsiveness, polite behavior, better physical facilities, and quick procedures as the key elements that determine the patient’s judgment about hospital quality [16,20,48]. A survey conducted in five countries of Central and Eastern Europe presented data on inpatient hospital satisfaction (or somewhat) with quality and accessibility, namely Bulgaria (85%; 86.8%), Hungary (93.1%; 93.5%), Lithuania (93.9%; 91.5%), Poland (87.4%; 79.7%), and Romania (87.8%; 86.2%), respectively. In particular, the mean scores of five dimensions (medical equipment at the hospital, reputation and skills of the physician/surgeon, condition of the hospital interior, attitude of the staff, and waiting time for the operation) were evaluated by service users on a scale between one and seven, as follows: Bulgaria (2.72, 1.49, 3.89, 3.99, and 5.32), Hungary (2.78, 1.93, 4.92, 3.70, and 4.04), Lithuania (2.41, 1.45, 5.04, 3.78, and 4.50), Poland (3.10, 2.00, 4.41, 4.19, and 4.06), and Romania (2.90, 1.95, 3.92, 3.84, and 5.09), respectively. The most important attribute appears to be ‘the reputation and skills of the physician/surgeon’ [49]. In a review aimed at describing patient experience and satisfaction data from countries with highly developed European health systems (Germany, Sweden, Finland, Norway, and the United Kingdom), ‘communication with physicians and nurses’ appeared to be an unequivocal factor for patient satisfaction in all countries, having a direct and positive impact on the overall hospital rating. The ability to communicate and provide useful information, including treatment guidelines, rights, obligations, ways to make complaints and suggestions, current health status, and processes after discharge, are crucial factors. Understaffing seems to be presented in many countries as a reason for dissatisfaction and can play an important role in patients’ hospital choice [50,51,52,53,54]. In Portugal and Slovenia, ‘waiting time’ was one of the most critical factors of patient satisfaction and an obstacle to hospital access; for this reason, patients’ dissatisfaction increases intensely when waiting time becomes more substantial, a factor that is usually connected with the organization in Emergency Medical Services (EMS) clinics [55,56,57,58,59]. Finally, safety was perceived as a fundamental indicator of patient satisfaction [60].

## 5. Conclusions

Patient orientation better ensures that the services provided are in line with patients’ needs and expectations. In several European countries, surveys are taking place to map the quality of care as perceived by patients in hospital settings, helping guide quality improvement actions and regulations. However, Greece has not yet established routines for measuring and publicizing patient experience and satisfaction, even though the WHO’s ‘World Health Report 2000’ recognizes that the quality of care perceived by patients (called ‘responsiveness’) has been seen as an integral part of the system’s performance.

Overall, this study revealed that the services provided by the participating hospitals were not entirely satisfactory to patients. The quality of medical services was rated as moderate, as patients’ satisfaction with their experience was rated at 67.38%. Notably, it was found that professionalism (>91%), courtesy (97.07%), discretion (97.46%), and respect (95.03%) for the patient’s personality were the factors that positively affected the patient’s satisfaction, which suggests that patients are more satisfied when they receive attention, kindness, and polite behavior from the healthcare providers. Even if both medical and nursing students are trained in patient interaction and the ‘bedside approach’, in the hectic demands of residency, priorities seem to shift to making a definitive diagnosis and treatment, sometimes at the expense of an empathic connection (44.42%) with the patient’s perspective. Some sociodemographic variables, such as gender, marital status, and nationality, were more consistent and relevant to patients’ perceptions.

The present study makes it possible to detect areas (those with the lowest ratings of satisfaction) for improvement in the healthcare service provided. More specifically, the following areas are highlighted: (1) accessibility, especially in the design of spaces and seating, waiting time planning (52%), and examination in emergencies. In addition, there appears to be delays in placing patients on wards, while in some hospitals, there is a problem with the provision of information at reception and with direction signs. (2) There are issues with scheduled procedures (68.05%) for the required examinations or operations, and the time of issuing the results. (3) In relation to security (53.29%), the safekeeping of personal belongings of patients and their companions is imperative. (4) In terms of tangibility, which is connected with the age of the buildings, there is a lack of availability and comfort of toilets, and improvements could be made in the technical condition of the equipment and the lack of materials (e.g., hygiene products; bed linen = 17.25%) in the wards and waiting areas. (5) There is room for improvement in the quality of food (64.10%). And last but not least (6), there are understaffing issues, especially severe shortages of physicians and nursing personnel, which affect responsiveness (51%), availability (64.57%), consultation time and the time they have for providing information (33.47%), communication, and the overall quality of care.

The following conclusions can be drawn from the results obtained. Effective communication, responsiveness, empathy, and clear explanations have the strongest impact on improving patient satisfaction and trust, among other characteristics of care. The public hospitals in Greece continue to be underfunded and lack strong support, while smart technologies and the digital transformation of data collection have not been adopted to help the low-paid staff to deliver better quality healthcare.

## 6. Strengths and Limitations

During the recruitment of participants, we did not register those patients who were unwilling or unable to participate in this study. Although the sample size is relatively large (3724 patients), we do not know if the excluded group differs from the sample. However, considering the heterogeneity, we do not expect major differences. Further, the findings of this survey refer to ten hospitals in the capital of Athens; therefore, it is not possible to draw firm conclusions about the whole population of Greek patients. This survey was fully based on public hospitals; private hospitals are not included here. In future, regarding the Aletras et al. [26] questionnaire, further research is required for its further validation with the use of larger samples from different hospitals.

## Figures and Tables

**Table 1 healthcare-12-00658-t001:** Sociodemographic characteristics of patients (Ν = 3724).

Characteristics	Frequency	Percentage
Gender		
Male	1802	48.39%
Female	1922	51.61%
Age (years)		
18–30	86	2.31%
31–44	347	9.32%
45–60	889	23.87%
61–74	1753	47.07%
>75	649	17.43%
Marital status		
Married	2755	73.98%
Single	209	5.61%
Divorced	223	5.99%
Widowed	537	14.42%
Education level		
Illiterate	17	0.46%
Primary school	366	9.83%
Secondary school	1114	29.91%
Compulsory	1617	43.42%
University	570	15.31%
Master or PhD	40	1.07%
Profession		
Farmer	43	1.15%
Civil servant	287	7.71%
Private employee	1392	37.38%
Retired	1423	38.21%
Worker	21	0.56%
Student	22	0.59%
Freelancer	293	7.87%
Unemployed or domestic	227	6.10%
Other	16	0.43%
Economic situation		
I cannot cope with my financial obligations	41	1.10%
I manage financially with great difficulties	2884	77.44%
I manage financially but I do not have much left aside	764	20.52%
I am financially comfortable	16	0.43%
I do not know or I do not answer	19	0.51%
Nationality		
Greek	2889	77.58%
Other	835	22.42%
Prefecture of residence		
Attica	3148	84.53%
Other	576	15.47%

**Table 2 healthcare-12-00658-t002:** The satisfaction rate and descriptive statistics of the admitted patients (Ν = 3724).

		Satisfied Patients *										
	Questions	N	%	Mean	SE **	Median	Variance	SD *	Min	Max	Range	IR **
Q8	The wait in a chair or wheelchair until they found you a bed in a ward was more than it should have been	1929	51.97%	2.81	0.190	2.00	1.730	1.315	1	5	4	2
Q10	Your admission to the Hospital was unpleasant	1311	35.28%	2.85	0.163	3.00	1.276	1.130	1	5	4	2
Q14	Nursing staff showed interest and took care of you during your hospitalization	3141	84.34%	3.98	0.131	4.00	0.829	0.911	2	5	3	1
Q15	Nursing staff were rude	3614	97.07%	4.02	0.135	4.00	0.872	0.934	1	5	4	1
Q16	Nursing staff explained the medical procedures (examinations by physicians. X-rays. etc.) in an understandable way.	1212	33.47%	3.63	0.167	4.00	1.346	1.160	1	5	4	2
Q17	Nursing staff were reluctant to answer your questions	2615	72.82%	3.85	0.143	4.00	0.978	0.989	1	5	4	0
Q18	Nursing staff discussed your concerns and fears with you	1617	44.42%	3.25	0.194	3.50	1.809	1.345	1	5	4	2
Q19	Nursing staff was slow to come when you asked for them	1856	50.99%	3.63	0.175	4.00	1.473	1.214	1	5	4	3
Q20	Nursing staff treated you with respect	3535	95.03%	4.06	0.117	4.00	0.656	0.810	1	5	4	1
Q22	Nursing staff were discreet	3549	96.91%	4.06	0.121	4.00	0.698	0.836	1	5	4	1
Q23	Doctors rarely stopped by to check on your health	3274	88.56%	3.69	0.137	4.00	0.900	0.949	1	5	4	1
Q26	Doctors have been keeping you informed about your health status	3509	94.23%	3.94	0.131	4.00	0.826	0.909	1	5	4	0
Q28	Doctors were discreet	3645	98.01%	4.00	0.123	4.00	0.723	0.851	1	5	4	0
Q30	The doctors were available when you want to ask them for something important	2404	64.57%	3.67	0.172	4.00	1.418	1.191	1	5	4	1
Q32	The hospital staff providing food services were courteous	3514	94.69%	4.06	0.124	4.00	0.741	0.861	1	5	4	1
Q33	The cleaners were rude	3380	91.92%	3.81	0.129	4.00	0.794	0.891	1	5	4	1
Q34	Other staff (e.g., admins, doormen, security staff, paramedics, canteen staff) were rude	2864	78.19%	3.52	0.136	4.00	0.893	0.945	1	5	4	1
Q38	Food quality was poor	2378	64.10%	2.88	0.167	3.00	1.346	1.160	1	5	4	2
Q41	The ward was clean	2967	79.87%	3.58	0.154	4.00	1.142	1.069	1	5	4	1
Q42	There was a commotion in the patient ward (room) where you were staying	2962	79.62%	3.00	0.179	3.00	1.532	1.238	1	5	4	2
Q43	Your bed sheets were changed as often as they should have been	636	17.25%	2.50	0.179	2.00	1.532	1.238	1	5	4	3
Q44	Room temperature was appropriate (cold or too hot)	3201	86.00%	3.23	0.166	4.00	1.329	1.153	1	5	4	2
Q45	The sanitary areas (bathrooms, toilets) were clean	2804	75.34%	3.06	0.182	4.00	1.592	1.262	1	5	4	2
Q46	In your ward you felt uncomfortable because there were more patients than there should have been	2877	77.32%	3.10	0.187	4.00	1.670	1.292	1	5	4	2
Q47	You felt that your personal belongings were adequately secured against potential theft	1968	53.29%	2.46	0.171	2.00	1.402	1.184	1	5	4	1
Q48	You felt uncomfortable because the relatives and friends of the patients stayed in the ward for a long time	2439	65.60%	2.98	0.164	3.00	1.297	1.139	1	5	4	2
Q49	The other areas of the hospital. e.g., (corridors. canteen. offices. waiting areas) were clean	2631	70.82%	3.54	0.163	4.00	1.275	1.129	1	5	4	1
Q51	Scheduled procedures (e.g., surgery. X-ray, blood, urine test, etc.) were performed without delay	2534	68.25%	3.29	0.160	4.00	1.232	1.110	1	5	4	2
Q52	Your hospital stay was disorganized and mistakes were made that could have been avoided (e.g., you were asked for your medical history again or an exam was repeated because it was lost)	3461	93.11%	3.71	0.146	4.00	1.020	1.010	1	5	4	0
Q56	Would you choose to come to the Hospital again?	3314	89.23%	3.56	0.149	4.00	1.060	1.029	1	5	4	1
Q57	Would you recommend the Hospital to your friends, family and third parties?	2565	69.16%	3.35	0.138	3.00	0.914	0.560	1	5	4	1

Notes: * in the two columns, the number and percentage of patients strongly agreeing or merely agreeing with a positive statement in each questionnaire item are presented. ** SE = standard error; SD = standard deviation; IR = interquartile range.

**Table 3 healthcare-12-00658-t003:** Dimensions associated with satisfaction of the admitted patients (Ν = 3724).

				Satisfied Patients *									
Dimensions		Questions	Number of Questions	N	%	Mean	SE **	Median	Variance	SD **	Min	Max	Range
1	Medical and nursing care	(Q 14–20, 22, 23, 26, 28, 30)	12	2657	81.13%	3.73	0.006	3.75	0.124	0.352	1.67	5.00	3.33
2	Organization and planning of hospitalization	(Q 8, 10, 51, 52)	4	1896	51.33%	3.35	0.010	3.50	0.374	0.611	1.00	5.00	4.00
3	Hospital environment	(Q 41–49)	9	1834	50.43%	3.44	0.008	3.56	0.209	0.457	1.00	5.00	4.00
4	Other quality factors	(Q 32, 33, 34, 38)	4	3121	86.65%	3.80	0.007	4.00	0.154	0.392	1.00	5.00	4.00
	Overall Satisfaction		29		67.38%								

Notes: * in the two columns, the number and percentage of patients strongly agreeing or merely agreeing with a positive statement in each questionnaire item are presented. ** SE = standard error; SD = standard deviation; positive ≥ 3.50; neutral = 2.50–3.49; negative = 0–2.49.

**Table 4 healthcare-12-00658-t004:** Multiple linear regression on dimensions of service quality for patient satisfaction.

			Patient Satisfaction			CI (95% Confidence Interval)	
Dimensions	B	SE	β	*t* Value	*p*	Lower Bound	Upper Bound
Medical and nursing care	0.285	0.027	0.175	10.469	<0.001	0.231	0.338
Organization and planning of hospitalization	0.198	0.016	0.215	12.752	<0.001	0.168	0.229
Hospital environment	0.264	0.020	0.215	13.134	<0.001	0.224	0.303
Other quality factors	0.295	0.023	0.207	12.847	<0.001	0.250	0.340
Constant	0.099	0.110		0.903	0.366	−0.116	0.315

Notes: R^2^ = 0.32; adjusted R^2^ = 0.32; SE = 0.46; F-value = 354.46; *p* = 0.00 and *p* ≤ 0.05.

**Table 5 healthcare-12-00658-t005:** Regression results determining the relationship between satisfaction and patients’ demographic characteristics and service quality dimensions.

	Patient Satisfaction Questionnaire Domains									
Variables	Medical and Nursing Care		Organization and Planning of Hospitalization		Hospital Environment		Other Quality Factors		Overall Satisfaction *	
	β Coef.	*p* Value	β Coef.	*p* Value	β Coef.	*p* Value	β Coef.	*p* Value	β Coef.	*p* Value
R^2^	0.002		0.000		0.003		0.001		0.006	
Gender										
Men vs. women	−0.043	<0.05	−0.003	0.855	−0.052	<0.05	−0.026	0.126	−0.075	<0.05
R^2^	0.002		0.022		0.024		0.006		0.017	
Marital status										
Married vs. other	−0.047	<0.05	−0.148	<0.05	−0.156	<0.05	−0.080	<0.05	−0.130	<0.05
R^2^	0.003		0.002		0.003		0.001		0.000	
Education level										
No studies/primary	-----	-----	-----	-----	-----	-----	-----	-----	-----	-----
Secondary	−0.057	<0.05	−0.043	<0.05	−0.050	<0.05	−0.028	0.091	−0.006	0.724
University/Master/PhD	0.129	<0.05	0.118	<0.05	−0.029	0.082	−0.007	0.694	−0.006	0.724
R^2^	0.006		0.005		0.000		0.001		0.000	
Economic situation										
Satisfied vs. other	−0.078	<0.05	−0.071	<0.05	−0.009	0.596	−0.024	0.152	0.004	0.812
R^2^	0.000		0.003		0.000		0.000		0.001	
Prefecture of residence										
Attica vs. other	0.004	0.800	−0.050	<0.05	0.009	0.579	0.010	0.553	0.031	0.062
R^2^	0.016		0.019		0.002		0.022		0.008	
Nationality										
Greek vs. other	−0.127	<0.05	−0.137	<0.05	0.040	<0.05	0.148	<0.05	0.089	<0.05

Notes: β Coef = beta coefficient from the lineal general model, after adjustment by all relevant variables. Positive values indicate more satisfaction with that domain for that category; negative values indicate less satisfaction compared with the reference category, which is blank or indicated as “versus”. * Grading scale = 0–10.

**Table 6 healthcare-12-00658-t006:** Regression analysis results of the key influencing factors related to preferring and recommending a hospital, and generally favorable thoughts about the services provided.

						CI (95% Confidence Interval)	
	B	SE	β	*t*	*p*	Lower Bound	Upper Bound
**Reasons for preferring the same hospital**							
Comfortable waiting area in the registration and admission office	0.024	0.010	0.044	2.494	0.013	0.005	0.043
Nursing staffs’ interest	0.100	0.016	0.110	6.386	0.000	0.069	0.130
Nursing staffs’ treatment	0.071	0.018	0.066	3.863	0.000	0.035	0.108
Physician’s time informing patients about their health status	0.071	0.018	0.065	3.924	0.000	0.035	0.106
Availability of physicians	0.091	0.018	0.082	5.009	0.000	0.055	0.126
Courtesy of cleaners	0.091	0.018	0.082	5.009	0.000	0.055	0.126
Cleanliness in the patient ward	−0.028	0.011	−0.041	−2.467	0.014	−0.050	−0.006
Food quality	0.054	0.010	0.092	5.508	0.000	0.035	0.073
Calmness in the patient ward	0.030	0.011	0.050	2.878	0.004	0.010	0.051
Appropriate temperature in the patient ward	0.047	0.012	0.064	3.976	0.000	0.024	0.070
Cleanliness in the sanitary areas	0.050	0.011	0.077	4.705	0.000	0.029	0.071
Security of personal belongings	0.052	0.009	0.094	5.560	0.000	0.034	0.070
Scheduled procedures without delays	0.131	0.011	0.213	12.197	0.000	0.110	0.153
Organization and planning of hospitalization	0.134	0.016	0.150	8.637	0.000	0.104	0.165
**Reasons for recommending the same hospital again**							
Comfortable waiting area in the registration and admission office	0.055	0.018	0.051	3.062	0.002	0.020	0.090
Βehavior of the nursing staff	0.046	0.018	0.041	2.558	0.011	0.011	0.082
Nursing staffs’ explanations about procedures	0.031	0.011	0.045	2.785	0.005	0.009	0.053
Empathy of the nursing staff	0.038	0.010	0.058	3.652	0.000	0.018	0.059
Physician’s time informing patients about their health status	0.056	0.021	0.044	2.702	0.007	0.015	0.096
Availability of physicians	0.058	0.010	0.089	5.537	0.000	0.037	0.078
Courtesy of cleaners	0.091	0.021	0.070	4.428	0.000	0.051	0.132
Food quality	0.053	0.011	0.076	4.715	0.000	0.031	0.074
Cleanliness in the sanitary areas	−0.030	0.013	−0.037	−2.321	0.020	−0.055	−0.005
Calmness in the patient ward	0.034	0.012	0.049	2.876	0.004	0.011	0.058
Changing of bed sheets	0.040	0.011	0.056	3.528	0.000	0.018	0.062
Appropriate temperature in the patient ward	0.055	0.013	0.063	4.051	0.000	0.028	0.081
Cleanliness in the sanitary areas	0.061	0.012	0.080	5.053	0.000	0.037	0.085
Security of personal belongings	0.067	0.011	0.102	6.233	0.000	0.046	0.088
Scheduled procedures without delays	0.203	0.012	0.279	16.495	0.000	0.179	0.227
Organization and planning of hospitalization	0.100	0.018	0.094	5.599	0.000	0.065	0.134
**Favorable acceptance of the healthcare services provided ***							
Shorter waiting period for admission	−0.183	0.021	−0.151	−8.696	0.000	−0.224	−0.142
Comfortable waiting area in the registration and admission office	0.136	0.023	0.097	5.819	0.000	0.090	0.182
Nursing staffs’ interest	0.233	0.038	0.102	6.184	0.000	0.159	0.307
Empathy of the nursing staff	0.100	0.022	0.071	4.516	0.000	0.057	0.144
Quick response of the nursing staff	−0.192	0.021	−0.149	−9.211	0.000	−0.233	−0.151
Physician’s time informing patients about their health status	0.117	0.043	0.043	2.698	0.007	0.032	0.202
Courtesy of staff providing food services	0.184	0.058	0.049	3.190	0.001	0.071	0.297
Courtesy of cleaners	0.185	0.043	0.067	4.248	0.000	0.099	0.270
Courtesy of other staff	0.076	0.032	0.036	2.348	0.019	0.013	0.140
Food quality	0.156	0.024	0.105	6.639	0.000	0.110	0.202
Cleanliness in the patient ward	−0.088	0.027	−0.051	−3.226	0.001	−0.142	−0.035
Calmness in the patient ward	0.081	0.025	0.054	3.218	0.001	0.032	0.131
Changing of bed sheets	0.068	0.024	0.045	2.846	0.004	0.021	0.114
Appropriate temperature in the patient ward	0.128	0.028	0.069	4.511	0.000	0.072	0.183
Scheduled procedures without delays	0.455	0.026	0.292	17.517	0.000	0.404	0.506
Organization and planning of hospitalization	0.293	0.038	0.130	7.796	0.000	0.219	0.367

Notes: B = unstandardized regression coefficient; SE = standard error; β = standardized regression coefficient; CI = confidence intervals. * Grading scale = 0–10.

## Data Availability

The data will be accessible from the corresponding author when the Ethics Committee of the National and Kapodistrian University of Athens and the 1st Regional Health Authority of Attica provide data access permission.

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
