# Peer review of "Exploring Inpatients’ Perspective: A Cross-Sectional Survey on Satisfaction and Experiences in Greek Hospitals"

_healthcare, 2024, doi:10.3390/healthcare12060658_

Round 1
Reviewer 1 Report
Comments and Suggestions for Authors
There seems to be a lack of research on patient satisfaction in the Greek healthcare system. Therefore, the study aims at filling an important gap. However, I think that in large parts of the manuscript, the reference to this specific context is missing and should be added. An improvement of the paper would result in embedding the results in information on the state and challenges of the healthcare system in Greece. In its current form, the results and their interpretation seem rather isolated.
Moreover, I would recommend that the authors review and revise the manuscript in terms of language/wording.
Please find my suggestions and comments below.
Abstract
In the "conclusion" passage, the authors should relate their findings to the specific healthcare context of Greece.
Introduction
p. 1, line 32: Please write "professionals" instead of "professionals' ".
p. 2, lines 64/65: I'm not sure whether this sentence is entirely correct - did the authors mean "hospital efforts to satisfy patients"? So please check if the comma after "satisfy" is correct.
p. 2, line 68: Please add "may" or "will" in the sentence ("A loyal patient may not only comply...").
What is missing at the end of the introduction in my view is the context - this study was conducted specifically in the Greek healthcare system. Therefore the authors should provide some information on what kinds of healthcare evaluation already exist, what gaps or deficits in health quality assurance can be found etc. This is necessary to put the research questions and results into perspective.
Materials and methods - instrument
p. 2, line 92: Please write "based on" instead of "based at".
p. 2, lines 94-96: This sentence is unclear, please check and revise. Generally, please revise the whole section in terms of language/wording.
Moreover, it would be helpful to have some further information about the questionnaire on which the instrument used in this study is based. I.e., what was that questionnaire developed for, ....?
Results
Please write just "results", not "research results".
p. 3, line 144: What is meant by "Kolmogorov-Smirnov test was developed"? Please consider deleting this part of the sentence.
p. 6: The information on hospitalization characteristics might be shortened and integrated into the sample characteristics section.
p. 6, line 175: How were the dimensions developed? For instance, were they derived statistically via factor analysis, were they defined by the authors, ...? Please provide information on this.
Table 2: How are the items sorted? By the dimensions mentioned before, in terms of their "satisfaction rate", ....? Please add information on this.
Tables 3 to 6: It is not very reader-friendly to simply list the tables without explanatory text in between that sums up the main findings described in the tables. Please add and revise. Then, section 3.9 (p. 12) is appropriate as a brief summary of the results.
Discussion
While it is generally appropriate that the authors discuss their study results in detail, I think they pay too much attention to details. As mentioned, the contextual element is missing: The "interesting findings" as stated by the authors (p. 12, lines 203 et seq.) refer to patient experiences in hospitals in a specific region in Greece. A central part of the discussion (and an interesting one for the readers) should rather be what these results mean for healthcare in Greece: Are they comparable to previous data? Is this the first comprehensive evaluation of patient satisfaction? If so, this can be added in the "strengths/limitations" section. What implications do the findings have for quality assurance programs or standards in the Greek healthcare system? And how do the results relate to findings from other countries/health contexts? The authors address these aspects rather briefly, for example, in the conclusions section (lack of standards/routines in Greece regarding the assessment of patient experience, p. 14, line 313). I would recommend that they put more emphasis in this whereas details on the results should be shortened.
Conclusions
p. 15, line 338: Shouldn't it be (9) instead of (10)?
p. 15, line 340: Please write "lack" instead of "lacks".
Author Response
Reviewer 1
Comments and Suggestions for Authors
There seems to be a lack of research on patient satisfaction in the Greek healthcare system. Therefore, the study aims at filling an important gap. However, I think that in large parts of the manuscript, the reference to this specific context is missing and should be added. An improvement of the paper would result in embedding the results in information on the state and challenges of the healthcare system in Greece. In its current form, the results and their interpretation seem rather isolated.
Moreover, I would recommend that the authors review and revise the manuscript in terms of language/wording.
Please find my suggestions and comments below.
Abstract
In the "conclusion" passage, the authors should relate their findings to the specific healthcare context of Greece.
Response: We followed the suggestion of reviewer. We remodeled the abstract in the “results” and "conclusion" passage to make the text relevant and more adequate to the specific health care context of Greece (page 1, lines 18-26)
Introduction
- 1, line 32: Please write "professionals" instead of "professionals' ".
Response: We followed the suggestion of reviewer.
- 2, lines 64/65: I'm not sure whether this sentence is entirely correct - did the authors mean "hospital efforts to satisfy patients"? So please check if the comma after "satisfy" is correct.
Response: We followed the suggestion of reviewer, we omitted the comma.
- 2, line 68: Please add "may" or "will" in the sentence ("A loyal patient may not only comply...").
Response: We followed the suggestion of reviewer.
What is missing at the end of the introduction in my view is the context - this study was conducted specifically in the Greek healthcare system. Therefore, the authors should provide some information on what kinds of healthcare evaluation already exist, what gaps or deficits in health quality assurance can be found etc. This is necessary to put the research questions and results into perspective.
Response: We followed the suggestion of reviewer. We added 13 lines (87-99) and 9 references (17-25) for support our manuscript.
Materials and methods - instrument
- 2, line 92: Please write "based on" instead of "based at".
Response: We followed the suggestion of reviewer.
- 2, lines 94-96: This sentence is unclear, please check and revise. Generally, please revise the whole section in terms of language/wording.
Response: We followed the suggestion of reviewer.
Moreover, it would be helpful to have some further information about the questionnaire on which the instrument used in this study is based. I.e., what was that questionnaire developed for, ....?
Response: We followed the suggestion of reviewer. We added 8 lines (103-110) for support our manuscript
Results
Please write just "results", not "research results".
Response: We followed the suggestion of reviewer.
- 3, line 144: What is meant by "Kolmogorov-Smirnov test was developed"? Please consider deleting this part of the sentence.
Response: We followed the suggestion of reviewer. We rephrase the sentence p. 4, lines 179-181. More information about data statistics we added in new section “2.4. Statistical Analysis”.
- 6: The information on hospitalization characteristics might be shortened and integrated into the sample characteristics section.
Response: We followed the suggestion of reviewer. The information on hospitalization characteristics shortened and integrated into the sample characteristics section. The new section was named: “3.2. Analysis of Sociodemographic and Hospitalization Characteristics of Study Participants”
- 6, line 175: How were the dimensions developed? For instance, were they derived statistically via factor analysis, were they defined by the authors, ...? Please provide information on this.
Response: We followed the suggestion of reviewer. Lines 103-112, 205-208 and 212-217 have information about the questionnaire, factor analysis and the dimensions.
Tables 3 to 6: It is not very reader-friendly to simply list the tables without explanatory text in between that sums up the main findings described in the tables. Please add and revise. Then, section 3.9 (p. 12) is appropriate as a brief summary of the results.
Response: We followed the suggestion of reviewer. We added lines 212-217, 221-226, 230-231 and 241-243 for be the passage more friendly.
Discussion
While it is generally appropriate that the authors discuss their study results in detail, I think they pay too much attention to details. As mentioned, the contextual element is missing: The "interesting findings" as stated by the authors (p. 12, lines 203 et seq.) refer to patient experiences in hospitals in a specific region in Greece. A central part of the discussion (and an interesting one for the readers) should rather be what these results mean for healthcare in Greece: Are they comparable to previous data? Is this the first comprehensive evaluation of patient satisfaction? If so, this can be added in the "strengths/limitations" section. What implications do the findings have for quality assurance programs or standards in the Greek healthcare system? And how do the results relate to findings from other countries/health contexts? The authors address these aspects rather briefly, for example, in the conclusions section (lack of standards/routines in Greece regarding the assessment of patient experience, p. 14, line 313). I would recommend that they put more emphasis in this whereas details on the results should be shortened.
Response: We followed the reviewer's suggestion. We have shortened the "results" section in the discussion section as much as possible, as many researchers and readers desire a more detailed analysis of the patient’s expectations (e.g. reviewer 3). We added in page 13, 11 lines (328-338) for the situation in Greece, comprehensive evaluation and comparatively with previous studies, also we added references (16,20,48). A connection with other European Health Systems already existed (p. 13 – lines 338-361. In section “conclusions” we formatted better the text into paragraphs and we added 10 lines (363-366 and 370-377). We believe that lines 366-369 give a clear message for the situation in Greece, lines 374-377 about the strengths of healthcare system, the disadvantages (lines 381-392) and what we suggested to Ministry of Health (lines 363-369 and 393-397). Of course, we formulated limitations section accordingly.
Conclusions
- 15, line 338: Shouldn't it be (9) instead of (10)?
Response: We followed the suggestion of reviewer.
- 15, line 340: Please write "lack" instead of "lacks".
Response: We followed the suggestion of reviewer.
Finally, we feel that it is necessary to thank you for your detailed instructions that helped us to improve and strengthen the presentation of this work. We think that we answered to every comment and we hope that we were able to meet your expectations.
Reviewer 2 Report
Comments and Suggestions for Authors
The authors evaluated patient satisfaction and experience in Greek hospitals in a cross-sectional manner. Overall the manuscript is well-written and this is an interesting study. I have a few comments that I think could help strengthen the presentation of the methods and results.
- In 2.3. statistical analysis, please provide details on what p-values were used to determine a statistical significance. For example, a two-sided p-value < 0.05 was used to determine statistical significance. Please elaborate on the regression analysis. What specific models were used? For example, multiple linear regression models were used to assess the association of patient demographic characteristics with patient satisfaction. What estimates were obtained from the regression analyses? For example, the point estimate of the beta coefficients and its associated 95% confidence intervals were obtained from the regression models.
- In line 145, please describe p-values in scientific notations if the p-values are very small. They will never equal 0. In lines 144-145, please provide more details on the K-S tests. Which variables are tested here for normality?
- You mentioned that a total of 3,724 patients were included in your study. Please provide more details on the recruitment of the study participants. For example, how many patients have you approached? How many agreed to participate in your study? What was the response rate? How many provided complete questionnaires? Were there any patients excluded from your study because of incomplete questionnaires or missing data?
- In line 351, you mentioned that you do not expect major difference between the patients who were willing to participate and those who did not. Could you provide data to support this statement? For example, if you don’t have the sociodemographic information on those who were unwilling to participate, you could compare the sociodemographics of your study participants to patients in Greek hospitals in general. Are there any large differences in the prevalence of different sociodemographic variables?
Author Response
Reviewer 2
Comments and Suggestions for Authors
The authors evaluated patient satisfaction and experience in Greek hospitals in a cross-sectional manner. Overall, the manuscript is well-written and this is an interesting study. I have a few comments that I think could help strengthen the presentation of the methods and results.
- In 2.3. statistical analysis, please provide details on what p-values were used to determine a statistical significance. For example, a two-sided p-value < 0.05 was used to determine statistical significance. Please elaborate on the regression analysis. What specific models were used? For example, multiple linear regression models were used to assess the association of patient demographic characteristics with patient satisfaction. What estimates were obtained from the regression analyses? For example, the point estimate of the beta coefficients and its associated 95% confidence intervals were obtained from the regression models.
Response: We followed the reviewer's suggestion. We added at page 4, 13 lines (163-175) with more statistical elements about the data and methods used
- In line 145, please describe p-values in scientific notations if the p-values are very small. They will never equal 0. In lines 144-145, please provide more details on the K-S tests. Which variables are tested here for normality?
Response: We followed the reviewer's suggestion. We added at page 4, 13 lines (163-175) with more statistical elements about the data and methods. Moreover, the section 2.3. Statistical Analysis, separated in two sections 2.3. Participants and Procedure and 2.4. Statistical Analysis.
- You mentioned that a total of 3,724 patients were included in your study. Please provide more details on the recruitment of the study participants. For example, how many patients have you approached? How many agreed to participate in your study? What was the response rate? How many provided complete questionnaires? Were there any patients excluded from your study because of incomplete questionnaires or missing data?
Response: We followed the reviewer's suggestion. We added at page 4, 13 lines (163-175) with more statistical elements about the data and methods. However, we can’t provide information about how many patients we have approached, we believe more than fourfold but we have no tangible evidence. We added more information from Greek Ministry of Health about population of patients in 10 hospitals of our survey at page 3 (lines 144-148). We shared 4,000 and we got back 3,724 questionnaires. The response rate was 93% (3,724/4,000), we added also this information (lines 184-185) in abstract. The individual's responses to the 'don't know/don't want to answer' options in the questionnaire were not included for the evaluation of acceptability, we added lines 169-170 and 3 references (26,29,30) for support our thesis. Finally, the section 2.3. Statistical Analysis, separated in two sections 2.3. Participants and Procedure and 2.4. Statistical Analysis.
- In line 351, you mentioned that you do not expect major difference between the patients who were willing to participate and those who did not. Could you provide data to support this statement? For example, if you don’t have the sociodemographic information on those who were unwilling to participate, you could compare the sociodemographics of your study participants to patients in Greek hospitals in general. Are there any large differences in the prevalence of different sociodemographic variables?
Response: We could not receive information about personal data of patients who were unwilling to participate in this survey. We provide data about patients of public hospitals in 2021 according the Greek Ministry of Health, but the ministry did not provide sociodemographic characteristics about patients (p. 3, line 144-148, reference 28). We think that the statement is formal and at the same time necessary for us and the journal. Also, our sample (3,724 patients) is too large, that’s why we mentioned that we do not expect major differences.
Finally, we feel that it is necessary to thank you for your detailed instructions that helped us to improve and strengthen the presentation of this work. We think that we answered to every comment and we hope that we were able to meet your expectations.
Reviewer 3 Report
Comments and Suggestions for Authors
1. There are too many errors overall. The table was created in a confusing manner using ‘,’ in all places where ‘.’ should have been used.
2. If we are looking at factors related to patient satisfaction in 10 hospitals using the existing research, it seems necessary to conduct a more detailed analysis. In particular, there may be characteristics of the hospital and the characteristics of the individuals who completed the survey, but I am not sure what implications an overall analysis that does not take into account may have.
3. In the tool, 1 point is strongly agree and 5 points are strongly disagree, so it is judged that it is not inconvenient to interpret it by calculating it backwards, as the higher the score, the more agree it is. At the same time, you will need to handle the 6 points as missing values well to see if there were any problems in calculating the average.
4. There is ‘satisfied patient’ in each table. What does this mean? Please describe the research methods and interpretation of what that means.
Author Response
Reviewer 3
Comments and Suggestions for Authors
- There are too many errors overall. The table was created in a confusing manner using ‘,’ in all places where ‘.’ should have been used.
Response: We followed the reviewer's suggestion. All errors refer to ‘,’ and ‘.’ were corrected.
- If we are looking at factors related to patient satisfaction in 10 hospitals using the existing research, it seems necessary to conduct a more detailed analysis. In particular, there may be characteristics of the hospital and the characteristics of the individuals who completed the survey, but I am not sure what implications an overall analysis that does not take into account may have.
Response: In our research we collected a number of 3,724 questionnaires from 10 different hospitals. According to the Greek Ministry of Health, the Attica region in total of 24 General Public Hospitals had developed 8,631 hospital beds and provided health services to 589,448 patients in 2021. In the same year, the ten hospitals we studied had developed 5,313 hospital beds (62% of total) and provided healthcare services to 392,425 patients (67% of the whole access population) (p.3, lines 143-148, reference 28). A structured and validated questionnaire was used (Aletras et al., 2009), appropriately adjusted to the needs of the study for gather information about patients' reported experiences. We used in total 65 questions (58 questions about patient satisfaction and 7 for socio-demographic data), 6 tables and more than 7 statistical analysis or methods (Anova, Kruskal-Wallis H, t-test, Mann-Whitney U, reliability analysis, regression results in determining the relationship between satisfaction and patients’ demographic characteristics and service quality dimensions and at least 4 Multiple linear regression models) were used.
- In the tool, 1 point is strongly agree and 5 points are strongly disagree, so it is judged that it is not inconvenient to interpret it by calculating it backwards, as the higher the score, the more agree it is. At the same time, you will need to handle the 6 points as missing values well to see if there were any problems in calculating the average.
Response: The questionnaire includes questions with statements, either positive or negative, so as to exclude positive bias. The final questionnaire, in total, includes 58 questions of which questions 2, 4, 6, 8, 10, 15, 17, 19, 21, 23, 25, 27, 29, 31, 33, 34, 38, 40, 42, 44, 46, 48, 50, 52 are reversed before the statistical analysis, since they are worded opposite to the others (p.3, lines 125-129). With the instructions of questionnaire, the individual's responses to the 'don't know/don't want to answer' options in the questionnaire must not be included for the evaluation of acceptability (p.4, lines 169-170 and references 26,29,30).
- There is ‘satisfied patient’ in each table. What does this mean? Please describe the research methods and interpretation of what that means.
Response: Table 2 & 3 contain a column named ‘satisfied patient’ with variables are presented as absolute (N) and relative (%) frequencies about patients’ satisfaction. There is already in notes (*) that describe that presented the opinions of patients that strongly agreeing or merely agreeing. All statistical methods that used described in section “2.4. Statistical Analysis” (p. 4, lines 162-176). Thank you very much about your comments.
Round 2
Reviewer 1 Report
Comments and Suggestions for Authors
Thank you for revising your manuscript, which is now easier to read and better structured. Some wording should still be corrected before publication, please see my comments below.
Introduction
line 88: Please write either "contributing to..." or "that contribute to...".
line 94/95: What is meant by the sentence "the healthcare system has been strongly hospital-centered due to the detection of significant pathogens presented in the primary care system"? Please consider revising this sentence or explaining it better.
Materials and methods
line 103: Please write "...was conducted with a sample...".
Results
line 208: Please write "...of the study to gather...".
lines 211-217: Please check and revise the newly added passages, as they are not quite grammatically correct.
Discussion
line 335: I would recommend to write "...recent studies" instead of "latest studies". Moreover, please write "focus on", not "focus in".
Conclusions
lines 370-374: Please check and revise the newly added passages, as they are not quite grammatically correct.
line 395: Please delete the comma following "Greece".
Author Response
Reviewer 1_Round 2
Open Review
(x) I would not like to sign my review report
( ) I would like to sign my review report Quality of English Language
(x) I am not qualified to assess the quality of English in this paper
( ) English very difficult to understand/incomprehensible
( ) Extensive editing of English language required
( ) Moderate editing of English language required
( ) Minor editing of English language required
( ) English language fine. No issues detected
|
Yes |
Can be improved |
Must be improved |
Not applicable |
|
|
Does the introduction provide sufficient background and include all relevant references? |
(x) |
( ) |
( ) |
( ) |
|
Are all the cited references relevant to the research? |
( ) |
( ) |
( ) |
(x) |
|
Is the research design appropriate? |
(x) |
( ) |
( ) |
( ) |
|
Are the methods adequately described? |
( ) |
(x) |
( ) |
( ) |
|
Are the results clearly presented? |
( ) |
(x) |
( ) |
( ) |
|
Are the conclusions supported by the results? |
( ) |
(x) |
( ) |
( ) |
Comments and Suggestions for Authors
Thank you for revising your manuscript, which is now easier to read and better structured. Some wording should still be corrected before publication, please see my comments below.
Introduction
line 88: Please write either "contributing to..." or "that contribute to...".
Response: We followed the suggestion of reviewer, we write "that contribute to..."
line 94/95: What is meant by the sentence "the healthcare system has been strongly hospital-centered due to the detection of significant pathogens presented in the primary care system"? Please consider revising this sentence or explaining it better.
Response: We followed the suggestion of reviewer and we revised the sentence appropriate, as to refer the pathogens in the primary care will change the structure of our manuscript.
Materials and methods
line 103: Please write "...was conducted with a sample...".
Response: We followed the suggestion of reviewer.
Results
line 208: Please write "...of the study to gather...".
Response: We followed the suggestion of reviewer.
lines 211-217: Please check and revise the newly added passages, as they are not quite grammatically correct.
Response: We followed the suggestion of reviewer.
Discussion
line 335: I would recommend to write "...recent studies" instead of "latest studies".
Moreover, please write "focus on", not "focus in".
Response: We followed the suggestion of reviewer,
Conclusions
lines 370-374: Please check and revise the newly added passages, as they are not quite grammatically correct.
Response: We followed the suggestion of reviewer
line 395: Please delete the comma following "Greece".
Response: We followed the suggestion of reviewer.
Moreover, reviewer notes:
Are all the cited references relevant to the research? = Not applicable
Response: We followed the suggestion of reviewer. References 21-25 & 32 upgraded. Now of the total of 60 references, 41 (68.33%) are between 2014-2024, 13 (21.67%) are between 2004-2013 and only 6 (10%) are before the year 2003.
Are the methods adequately described? = Can be improved
Response: We followed the suggestion of reviewer. Improvements in lines 206-207, 213-218, 222-229, 233-235 were carried out.
Are the conclusions supported by the results? = Can be improved
Response: We followed the suggestion of reviewer. Improvements in lines 375-400 were carried out, so as conclusions supported by the results
Some wording and phrases upgraded, lines 82, 84-85, 93-96, 172, 185, 243, 275, 340-341.
Finally, we feel that it is necessary to thank you for your detailed instructions that helped us to improve and strengthen the presentation of this work. We think that we answered to every comment and we hope that we were able to meet your expectations.